# Approximate Dynamic Programming Finally Performs Well in the Game of Tetris

**Victor Gabillon**
INRIA Lille - Nord Europe,
Team SequeL, FRANCE
*victor.gabillon@inria.fr*

**Mohammad Ghavamzadeh**[*]
INRIA Lille - Team SequeL
& Adobe Research
*mohammad.ghavamzadeh@inria.fr*

**Bruno Scherrer**
INRIA Nancy - Grand Est,
Team Maia, FRANCE
*bruno.scherrer@inria.fr*

## Abstract

Tetris is a video game that has been widely used as a benchmark for various optimization techniques including approximate dynamic programming (ADP) algorithms. A look at the literature of this game shows that while ADP algorithms that have been (almost) entirely based on approximating the value function (value function based) have performed poorly in Tetris, the methods that search directly in the space of policies by learning the policy parameters using an optimization black box, such as the cross entropy (CE) method, have achieved the best reported results. This makes us conjecture that Tetris is a game in which good policies are easier to represent, and thus, learn than their corresponding value functions. So, in order to obtain a good performance with ADP, we should use ADP algorithms that search in a policy space, instead of the more traditional ones that search in a value function space. In this paper, we put our conjecture to test by applying such an ADP algorithm, called classification-based modified policy iteration (CBMPI), to the game of Tetris. Our experimental results show that for the first time an ADP algorithm, namely CBMPI, obtains the best results reported in the literature for Tetris in both small $10 \times 10$ and large $10 \times 20$ boards. Although the CBMPI's results are similar to those of the CE method in the large board, CBMPI uses considerably fewer (almost 1/6) samples (calls to the generative model) than CE.

## 1 Introduction

Tetris is a popular video game created by Alexey Pajitnov in $1985$. The game is played on a grid originally composed of 20 rows and 10 columns, where pieces of 7 different shapes fall from the top – see Figure 1. The player has to choose where to place each falling piece by moving it horizontally and rotating it. When a row is filled, it is removed and all the cells above it move one line down. The goal is to remove as many rows as possible before the game is over, i.e., when there is no space available at the top of the grid for the new piece.

In this paper, we consider the variation of the game in which the player knows only the current falling piece, and not the next several coming pieces. This game constitutes an interesting optimization benchmark in which the goal is to find a controller (policy) that maximizes the average (over multiple games) number of lines removed in a game (score).[1] This optimization problem is known to be computationally hard. It contains a huge number of board configurations (about $2^{200} \simeq 1.6 \times 10^{60}$), and even in the case that the sequence of pieces is known in advance, finding the optimal strategy is an NP hard problem [4].

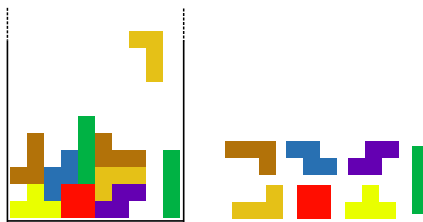

Figure 1: A screen-shot of the game of Tetris with its seven pieces (shapes).

Approximate dynamic programming (ADP) and reinforcement learning (RL) algorithms have been used in Tetris. These algorithms formulate Tetris as a Markov decision process (MDP) in which the state is defined by the current board configuration plus the falling piece, the actions are the

---

[*]Mohammad Ghavamzadeh is currently at Adobe Research, on leave of absence from INRIA.

[1]Note that this number is finite because it was shown that Tetris is a game that ends with probability one [3].

possible orientations of the piece and the possible locations that it can be placed on the board,[2] and the reward is defined such that maximizing the expected sum of rewards from each state coincides with maximizing the score from that state. Since the state space is large in Tetris, these methods use value function approximation schemes (often linear approximation) and try to tune the value function parameters (weights) from game simulations. The first application of ADP in Tetris seems to be by Tsitsiklis and Van Roy [22]. They used the approximate value iteration algorithm with two state features: the board height and the number of holes in the board, and obtained a low score of 30 to 40. Bertsekas and Ioffe [1] proposed the $\lambda$-Policy Iteration ($\lambda$-PI) algorithm (a generalization of value and policy iteration) and applied it to Tetris. They approximated the value function as a linear combination of a more elaborate set of 22 features and reported the score of 3, 200 lines. The exact same empirical study was revisited recently by Scherrer [16], who corrected an implementation bug in [1], and reported more stable learning curves and the score of 4, 000 lines. At least three other ADP and RL papers have used the same set of features, we refer to them as the "Bertsekas features", in the game of Tetris. Kakade [11] applied a natural policy gradient method to Tetris and reported a score of about 6, 800 lines. Farias and Van Roy [6] applied a linear programming algorithm to the game and achieved the score of 4, 700 lines. Furmston and Barber [8] proposed an approximate Newton method to search in a policy space and were able to obtain a score of about 14, 000.

Despite all the above applications of ADP in Tetris (and possibly more), for a long time, the best Tetris controller was the one designed by Dellacherie [5]. He used a heuristic evaluation function to give a score to each possible strategy (in a way similar to value function in ADP), and eventually returned the one with the highest score. Dellacherie's evaluation function is made of 6 high-quality features with weights chosen by hand, and achieved a score of about 5, 000, 000 lines [19]. Szita and Lőrincz [18] used the "Bertsekas features" and optimized the weights by running a black box optimizer based on the cross entropy (CE) method [15]. They reported the score of 350, 000 lines averaged over 30 games, outperforming the ADP and RL approaches that used the same features. More recently, Thiery and Scherrer [20] selected a set of 9 features (including those of Dellacherie's) and optimized the weights with the CE method. This led to the best publicly known controller (to the best of our knowledge) with the score of around 35, 000, 000 lines.

Due to the high variance of the score and its sensitivity to some implementation details [19], it is difficult to have a precise evaluation of Tetris controllers. However, our brief tour d'horizon of the literature, and in particular the work by Szita and Lőrincz [18] (optimizing the "Bertsekas features" by CE), indicate that ADP algorithms, even with relatively good features, have performed extremely worse than the methods that directly search in the space of policies (such as CE and genetic algorithms). It is important to note that almost all these ADP methods are *value function based* algorithms that first define a value function representation (space) and then search in this space for a good function, which later gives us a policy.

The main motivation of our work comes from the above observation. This observation makes us conjecture that Tetris is a game whose policy space is easier to represent, and as a result to search in, than its value function space. Therefore, in order to obtain a good performance with ADP algorithms in this game, we should use those ADP methods that search in a policy space, instead of the more traditional ones that search in a value function space. Fortunately a class of such ADP algorithms, called *classification-based policy iteration* (CbPI), have been recently developed and analyzed [12, 7, 13, 9, 17]. These algorithms differ from the standard *value function based ADP* methods in how the greedy policy is computed. Specifically, at each iteration CbPI algorithms approximate the entire greedy policy as the output of a classifier, while in the standard methods, at every given state, the required action from the greedy policy is individually calculated based on the approximation of the value function of the current policy. Since CbPI methods search in a policy space (defined by a classifier) instead of a value function space, we believe that they should perform better than their value function based counterparts in problems in which good policies are easier to represent than their corresponding value functions. In this paper, we put our conjecture to test by applying an algorithm in this class, called *classification-based modified policy iteration* (CBMPI) [17], to the game of Tetris, and compare its performance with the CE method and the $\lambda$-PI algorithm. The choice of CE and $\lambda$-PI is because the former has achieved the best known results in Tetris and the latter's performance is among the best reported for value function based ADP algorithms. Our extensive experimental results show that for the first time an ADP algorithm, namely CBMPI, obtains the best results reported in the literature for Tetris in both small $10 \times 10$ and large $10 \times 20$ boards. Although

Figure 2: The pseudo-code of the cross-entropy (CE) method used in our experiments.

the CBMPI's results are similar to those achieved by the CE method in the large board, CBMPI uses considerably fewer (almost 1/6) samples (call to the generative model of the game) than CE. In Section 2, we briefly describe the algorithms used in our experiments. In Section 3, we outline the setting of each algorithm in our experiments and report our results followed by discussion.

## 2 Algorithms

In this section, we briefly describe the algorithms used in our experiments: the cross entropy (CE) method, classification-based modified policy iteration (CBMPI) [17] and its slight variation direct policy iteration (DPI) [13], and $\lambda$-policy iteration (see [16] for a description of $\lambda$-PI). We begin by defining some terms and notations. A state $s$ in Tetris consists of two components: the description of the board $b$ and the type of the falling piece $p$. All controllers rely on an evaluation function that gives a value to each possible action at a given state. Then, the controller chooses the action with the highest value. In ADP, algorithms aim at tuning the weights such that the evaluation function approximates well the optimal expected future score from each state. Since the total number of states is large in Tetris, the evaluation function $f$ is usually defined as a linear combination of a set of features $\phi$, i.e., $f(\cdot) = \phi(\cdot)\theta$. We can think of the parameter vector $\theta$ as a policy (controller) whose performance is specified by the corresponding evaluation function $f(\cdot) = \phi(\cdot)\theta$. The features used in Tetris for a state-action pair $(s, a)$ may depend on the description of the board $b'$ resulted from taking action $a$ in state $s$, e.g., the maximum height of $b'$. Computing such features requires the knowledge of the game's dynamics, which is known in Tetris.

### 2.1 Cross Entropy Method

Cross-entropy (CE) [15] is an iterative method whose goal is to optimize a function $f$ parameterized by a vector $\theta \in \Theta$ by direct search in the parameter space $\Theta$. Figure 2 contains the pseudo-code of the CE algorithm used in our experiments [18, 20]. At each iteration $k$, we sample $n$ parameter vectors $\{\theta_i\}_{i=1}^n$ from a multivariate Gaussian distribution $\mathcal{N}(\mu, \sigma^2 I)$. At the beginning, the parameters of this Gaussian have been set to cover a wide region of $\Theta$. For each parameter $\theta_i$, we play $L$ games and calculate the average number of rows removed by this controller (an estimate of the evaluation function). We then select $\lfloor \rho n \rfloor$ of these parameters with the highest score, $\theta_1', \ldots, \theta_{\lfloor \rho n \rfloor}'$, and use them to update the mean $\mu$ and variance $\sigma^2$ of the Gaussian distribution, as shown in Figure 2. This updated Gaussian is used to sample the $n$ parameters at the next iteration. The goal of this update is to sample more parameters from the promising part of $\Theta$ at the next iteration, and eventually converge to a global maximum of $f$.

### 2.2 Classification-based Modified Policy Iteration (CBMPI)

Modified policy iteration (MPI) [14] is an iterative algorithm to compute the optimal policy of a MDP that starts with initial policy $\pi_1$ and value $v_0$, and generates a sequence of value-policy pairs

$$v_k = (T_{\pi_k})^m v_{k-1} \quad \text{(evaluation step)}, \qquad \pi_{k+1} = \mathcal{G}\big[(T_{\pi_k})^m v_{k-1}\big] \quad \text{(greedy step)},$$

where $\mathcal{G}v_k$ is a *greedy* policy w.r.t. $v_k$, $T_{\pi_k}$ is the Bellman operator associated with the policy $\pi_k$, and $m \geq 1$ is a parameter. MPI generalizes the well-known value and policy iteration algorithms for the values $m = 1$ and $m = \infty$, respectively. CBMPI [17] is an approximation of MPI that uses an explicit representation for the policies $\pi_k$, in addition to the one used for the value functions $v_k$. The idea is similar to the classification-based PI algorithms [12, 7, 13] in which we search for the greedy policy in a policy space $\Pi$ (defined by a classifier) instead of computing it from the estimated value function (as in the standard implementation of MPI). As described in Figure 3, CBMPI begins with an arbitrary initial policy $\pi_1 \in \Pi$ and value function $v_0 \in \mathcal{F}$.[3] At each iteration $k$, a new value func-

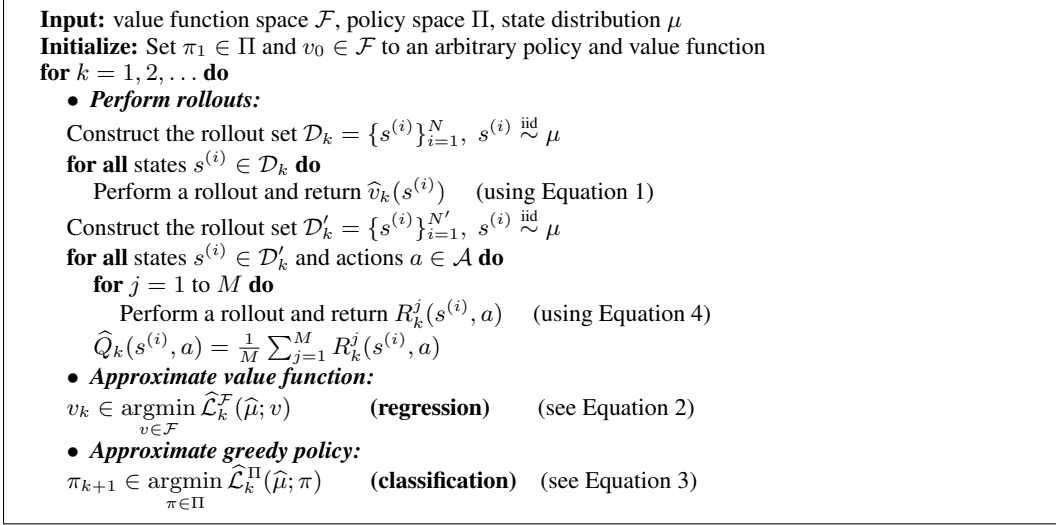

**Input:** value function space $\mathcal{F}$, policy space $\Pi$, state distribution $\mu$
**Initialize:** Set $\pi_1 \in \Pi$ and $v_0 \in \mathcal{F}$ to an arbitrary policy and value function
**for** $k = 1, 2, \ldots$ **do**
   • *Perform rollouts:*
   Construct the rollout set $\mathcal{D}_k = \{s^{(i)}\}_{i=1}^N$, $s^{(i)} \stackrel{\text{iid}}{\sim} \mu$
   **for all** states $s^{(i)} \in \mathcal{D}_k$ **do**
      Perform a rollout and return $\widehat{v}_k(s^{(i)})$   (using Equation 1)
   Construct the rollout set $\mathcal{D}'_k = \{s^{(i)}\}_{i=1}^{N'}$, $s^{(i)} \stackrel{\text{iid}}{\sim} \mu$
   **for all** states $s^{(i)} \in \mathcal{D}'_k$ and actions $a \in \mathcal{A}$ **do**
      **for** $j = 1$ to $M$ **do**
         Perform a rollout and return $R_k^j(s^{(i)}, a)$   (using Equation 4)
      $\widehat{Q}_k(s^{(i)}, a) = \frac{1}{M} \sum_{j=1}^M R_k^j(s^{(i)}, a)$
   • *Approximate value function:*
   $v_k \in \underset{v \in \mathcal{F}}{\arg\min} \, \widehat{\mathcal{L}}_k^{\mathcal{F}}(\widehat{\mu}; v)$      **(regression)**     (see Equation 2)
   • *Approximate greedy policy:*
   $\pi_{k+1} \in \underset{\pi \in \Pi}{\arg\min} \, \widehat{\mathcal{L}}_k^{\Pi}(\widehat{\mu}; \pi)$    **(classification)**   (see Equation 3)

Figure 3: The pseudo-code of the CBMPI algorithm.

tion $v_k$ is built as the best approximation of the $m$-step Bellman operator $(T_{\pi_k})^m v_{k-1}$ in $\mathcal{F}$ (*evaluation step*). This is done by solving a regression problem whose target function is $(T_{\pi_k})^m v_{k-1}$. To set up the regression problem, we build a rollout set $\mathcal{D}_k$ by sampling $N$ states i.i.d. from a distribution $\mu$. For each state $s^{(i)} \in \mathcal{D}_k$, we generate a rollout $\left(s^{(i)}, a_0^{(i)}, r_0^{(i)}, s_1^{(i)}, \ldots, a_{m-1}^{(i)}, r_{m-1}^{(i)}, s_m^{(i)}\right)$ of size $m$, where $a_t^{(i)} = \pi_k(s_t^{(i)})$, and $r_t^{(i)}$ and $s_{t+1}^{(i)}$ are the reward and next state induced by this choice of action. From this rollout, we compute an unbiased estimate $\widehat{v}_k(s^{(i)})$ of $\left[(T_{\pi_k})^m v_{k-1}\right](s^{(i)})$ as

$$\widehat{v}_k(s^{(i)}) = \sum_{t=0}^{m-1} \gamma^t r_t^{(i)} + \gamma^m v_{k-1}(s_m^{(i)}), \qquad (\gamma \text{ is the discount factor}), \tag{1}$$

and use it to build a training set $\left\{\left(s^{(i)}, \widehat{v}_k(s^{(i)})\right)\right\}_{i=1}^N$. This training set is then used by the regressor to compute $v_k$ as an estimate of $(T_{\pi_k})^m v_{k-1}$. The regressor finds a function $v \in \mathcal{F}$ that minimizes the empirical error

$$\widehat{\mathcal{L}}_k^{\mathcal{F}}(\widehat{\mu}; v) = \frac{1}{N} \sum_{i=1}^N \left(\widehat{v}_k(s^{(i)}) - v(s^{(i)})\right)^2. \tag{2}$$

The *greedy step* at iteration $k$ computes the policy $\pi_{k+1}$ as the best approximation of $\mathcal{G}\left[(T_{\pi_k})^m v_{k-1}\right]$ by minimizing the cost-sensitive empirical error (cost-sensitive classification)

$$\widehat{\mathcal{L}}_k^{\Pi}(\widehat{\mu}; \pi) = \frac{1}{N'} \sum_{i=1}^{N'} \left[ \max_{a \in \mathcal{A}} \widehat{Q}_k(s^{(i)}, a) - \widehat{Q}_k\left(s^{(i)}, \pi(s^{(i)})\right) \right]. \tag{3}$$

To set up this cost-sensitive classification problem, we build a rollout set $\mathcal{D}'_k$ by sampling $N'$ states i.i.d. from a distribution $\mu$. For each state $s^{(i)} \in \mathcal{D}'_k$ and each action $a \in \mathcal{A}$, we build $M$ independent rollouts of size $m+1$, i.e., $\left(s^{(i)}, a, r_0^{(i,j)}, s_1^{(i,j)}, a_1^{(i,j)}, \ldots, a_m^{(i,j)}, r_m^{(i,j)}, s_{m+1}^{(i,j)}\right)_{j=1}^M$, where for $t \geq 1$, $a_t^{(i,j)} = \pi_k(s_t^{(i,j)})$, and $r_t^{(i,j)}$ and $s_{t+1}^{(i,j)}$ are the reward and next state induced by this choice of action. From these rollouts, we compute an unbiased estimate of $Q_k(s^{(i)}, a)$ as $\widehat{Q}_k(s^{(i)}, a) = \frac{1}{M} \sum_{j=1}^M R_k^j(s^{(i)}, a)$ where each rollout estimate is defined as

$$R_k^j(s^{(i)}, a) = \sum_{t=0}^m \gamma^t r_t^{(i,j)} + \gamma^{m+1} v_{k-1}(s_{m+1}^{(i,j)}). \tag{4}$$

If we remove the regressor from CBMPI and only use the $m$-truncated rollouts $R_k^j(s^{(i)}, a) = \sum_{t=0}^m \gamma^t r_t^{(i,j)}$ to compute $\widehat{Q}_k(s^{(i)}, a)$, then CBMPI become the direct policy iteration (DPI) algorithm [13] that we also use in our experiments (see [17] for more details on the CBMPI algorithm).

**In our implementation of CBMPI (DPI) in Tetris** (Section 3), we use the same rollout set ($\mathcal{D}_k = \mathcal{D}'_k$) and rollouts for the classifier and regressor. This is mainly to be more sample efficient. Fortunately, we observed that this does not affect the overall performance of the algorithm. We set the discount factor $\gamma = 1$. **Regressor:** We use linear function approximation for the value function, i.e., $\widehat{v}_k(s^{(i)}) = \phi(s^{(i)})w$, where $\phi(\cdot)$ and $w$ are the feature and weight vectors, and minimize the empirical error $\widehat{\mathcal{L}}_k^{\mathcal{F}}(\widehat{\mu}; v)$ using the standard least-squares method. **Classifier:** The training set of the classifier is of size $N$ with $s^{(i)} \in \mathcal{D}'_k$ as input and $\left(\max_a \widehat{Q}_k(s^{(i)}, a) - \widehat{Q}_k(s^{(i)}, a_1), \ldots, \max_a \widehat{Q}_k(s^{(i)}, a) - \widehat{Q}_k(s^{(i)}, a_{|\mathcal{A}|})\right)$ as output. We use the policies of the form $\pi_u(s) = \operatorname{argmax}_a \psi(s, a)u$, where $\psi$ is the policy feature vector (possibly different from the value function feature vector $\phi$) and $u$ is the policy parameter vector. We compute the next policy $\pi_{k+1}$ by minimizing the empirical error $\widehat{\mathcal{L}}_k^{\Pi}(\widehat{\mu}; \pi_u)$, defined by (3), using the covariance matrix adaptation evolution strategy (CMA-ES) algorithm [10]. In order to evaluate a policy $u$ in CMA-ES, we only need to compute $\widehat{\mathcal{L}}_k^{\Pi}(\widehat{\mu}; \pi_u)$, and given the training set, this procedure does not require any simulation of the game. This is in contrary with policy evaluation in CE that requires playing several games, and it is the main reason that we obtain the same performance as CE with CBMPI with almost $1/6$ number of samples (see Section 3.2).

# 3 Experimental Results

In this section, we evaluate the performance of CBMPI (DPI) and compare it with CE and $\lambda$-PI. CE is the state-of-the-art method in Tetris with huge performance advantage over ADP/RL methods [18, 19, 20]. In our experiments, we show that for a well-selected set of features, CBMPI improves over all the previously reported ADP results. Moreover, its performance is comparable to that of the CE method, while using considerably fewer samples (call to the generative model of the game).

## 3.1 Experimental Setup

In our experiments, the policies learned by the algorithms are evaluated by their score (average number of rows removed in a game) averaged over 200 games in the small $10 \times 10$ board and over 20 games in the large $10 \times 20$ board. The performance of each algorithm is represented by a learning curve whose value at each iteration is the average score of the policies learned by the algorithm at that iteration in 100 separate runs of the algorithm. In addition to their score, we also evaluate the algorithms by the number of samples they use. In particular, we show that CBMPI/DPI use 6 times less samples than CE. As discussed in Section 2.2, this is due the fact that although the classifier in CBMPI/DPI uses a direct search in the space of policies (for the greedy policy), it evaluates each candidate policy using the empirical error of Eq. 3, and thus, does not require any simulation of the game (other than those used to estimate the $\widehat{Q}_k$'s in its training set). In fact, the budget $B$ of CBMPI/DPI is fixed in advance by the number of rollouts $NM$ and the rollout's length $m$ as $B = (m+1)NM|\mathcal{A}|$. In contrary, CE evaluates a candidate policy by playing several games, a process that can be extremely costly (sample-wise), especially for good policies in the large board.

**In our CBMPI/DPI experiments**, we set the number of rollouts per state-action pair $M = 1$, as this value has shown the best performance. Thus, we only study the behavior of CBMPI/DPI as a function of $m$ and $N$. In CBMPI, the parameter $m$ balances between the errors in evaluating the value function and the policy. For large values of $m$, the size of the rollout set decreases as $N = O(B/m)$, which in turn decreases the accuracy of both the regressor and classifier. This leads to a trade-off between long rollouts and the number of states in the rollout set. The solution to this trade-off (bias/variance tradeoff in estimation of $\widehat{Q}_k$'s) strictly depends on the capacity of the value function space $\mathcal{F}$. A rich value function space leads to solve the trade-off for small values of $m$, while a poor space, or no space in the case of DPI, suggests large values of $m$, but not too large to still guarantee a large enough $N$. We sample the rollout states in CBMPI/DPI from the trajectories generated by a very good policy for Tetris, namely the DU controller [20]. Since the DU policy is good, this rollout set is biased towards boards with small height. We noticed from our experiments that the performance can be significantly improved if we use boards with different heights in the rollout sets. This means that better performance can be achieved with more uniform sampling distribution, which is consistent with what we can learn from the CBMPI and DPI performance bounds. We set the initial value function parameter to $w = \bar{0}$ and select the initial policy $\pi_1$ (policy parameter $u$) randomly. We also set the CMA-ES parameters (classifier parameters) to $\rho = 0.5$, $\eta = 0$, and $n$ equal to 15 times the number of features.

**In the CE experiments**, we set $\rho = 0.1$ and $\eta = 4$, the best parameters reported in [20]. We also set $n = 1000$ and $L = 10$ in the small board and $n = 100$ and $L = 1$ in the large board.

**Set of Features:** We use the following features, plus a constant offset feature, in our experiments:[4]
*(i) Bertsekas features:* First introduced by [2], this set of 22 features has been mainly used in the ADP/RL community and consists of: the number of holes in the board, the height of each column, the difference in height between two consecutive columns, and the maximum height of the board.
*(ii) Dellacherie-Thiery (D-T) features:* This set consists of the six features of Dellacherie [5], i.e., the landing height of the falling piece, the number of eroded piece cells, the row transitions, the column transitions, the number of holes, and the number of board wells; plus 3 additional features proposed in [20], i.e., the hole depth, the number of rows with holes, and the pattern diversity feature. Note that the best policies reported in the literature have been learned using this set of features.
*(iii) RBF height features:* These new 5 features are defined as $\exp(\frac{-|c-ih/4|^2}{2(h/5)^2})$, $i = 0, \ldots, 4$, where $c$ is the average height of the columns and $h = 10$ or $20$ is the total number of rows in the board.

### 3.2 Experiments

We first run the algorithms on the small board to study the role of their parameters and to select the best features and parameters (Section 3.2.1). We then use the selected features and parameters and apply the algorithms to the large board (Figure 5 (d)) Finally, we compare the best policies found in our experiments with the best controllers reported in the literature (Tables 1 and 2).

#### 3.2.1 Small ($10 \times 10$) Board

Here we run the algorithms with two different feature sets: *Dellacherie-Thiery (D-T)* and *Bertsekas*.

**D-T features:** Figure 4 shows the learning curves of CE, $\lambda$-PI, DPI, and CBMPI algorithms. Here we use D-T features for the evaluation function in CE, the value function in $\lambda$-PI, and the policy in DPI and CBMPI. We ran CBMPI with different feature sets for the value function and "D-T plus the 5 RBF features" achieved the best performance (Figure 4 (d)).[5] The budget of CBMPI and DPI is set to $B = 8,000,000$ per iteration. The CE method reaches the score $3000$ after $10$ iterations using an average budget $\bar{B} = 65,000,000$. $\lambda$-PI with the best value of $\lambda$ only manages to score $400$. In Figure 4 (c), we report the performance of DPI for different values of $m$. DPI achieves its best performance for $m = 5$ and $m = 10$ by removing $3400$ lines on average. As explained in Section 3.1, having short rollouts ($m = 1$) in DPI leads to poor action-value estimates $\widehat{Q}$, while having too long rollouts ($m = 20$) decreases the size of the training set of the classifier $N$. CBMPI outperforms the other algorithms, including CE, by reaching the score of $4300$ for $m = 2$. The value of $m = 2$ corresponds to $N = \frac{8000000}{(2+1)\times 32} \approx 84,000$. Note that unlike DPI, CBMPI achieves good performance with very short rollouts $m = 1$. This indicates that CBMPI is able to approximate the value function well, and as a result, to build a more accurate training set for its classifier than DPI. The results of Figure 4 show that an ADP algorithm, namely CBMPI, outperforms the CE method using a similar budget ($80$ vs. $65$ millions after $10$ iterations). Note that CBMPI takes less iterations to converge than CE. More generally Figure 4 confirms the superiority of the policy search and classification-based PI methods to value function based ADP algorithms ($\lambda$-PI). This suggests that the D-T features are more suitable to represent the policies than the value functions in Tetris.

**Bertsekas features:** Figures 5 (a)-(c) show the performance of CE, $\lambda$-PI, DPI, and CBMPI algorithms. Here all the approximations in the algorithms are with the Bertsekas features. CE achieves the score of $500$ after about $60$ iterations and outperforms $\lambda$-PI with score of $350$. It is clear that the Bertsekas features lead to much weaker results than those obtained by the D-T features in Figure 4 for all the algorithms. We may conclude then that the D-T features are more suitable than the Bertsekas features to represent both value functions and policies in Tetris. In DPI and CBMPI, we managed to obtain results similar to CE, only after multiplying the per iteration budget $B$ used in the D-T experiments by $10$. However, CBMPI and CE use the same number of samples, $150,000,000$, when they converge after $2$ and $60$ iterations, respectively (see Figure 5). Note that DPI and CBMPI obtain the same performance, which means that the use of a value function approximation by CBMPI

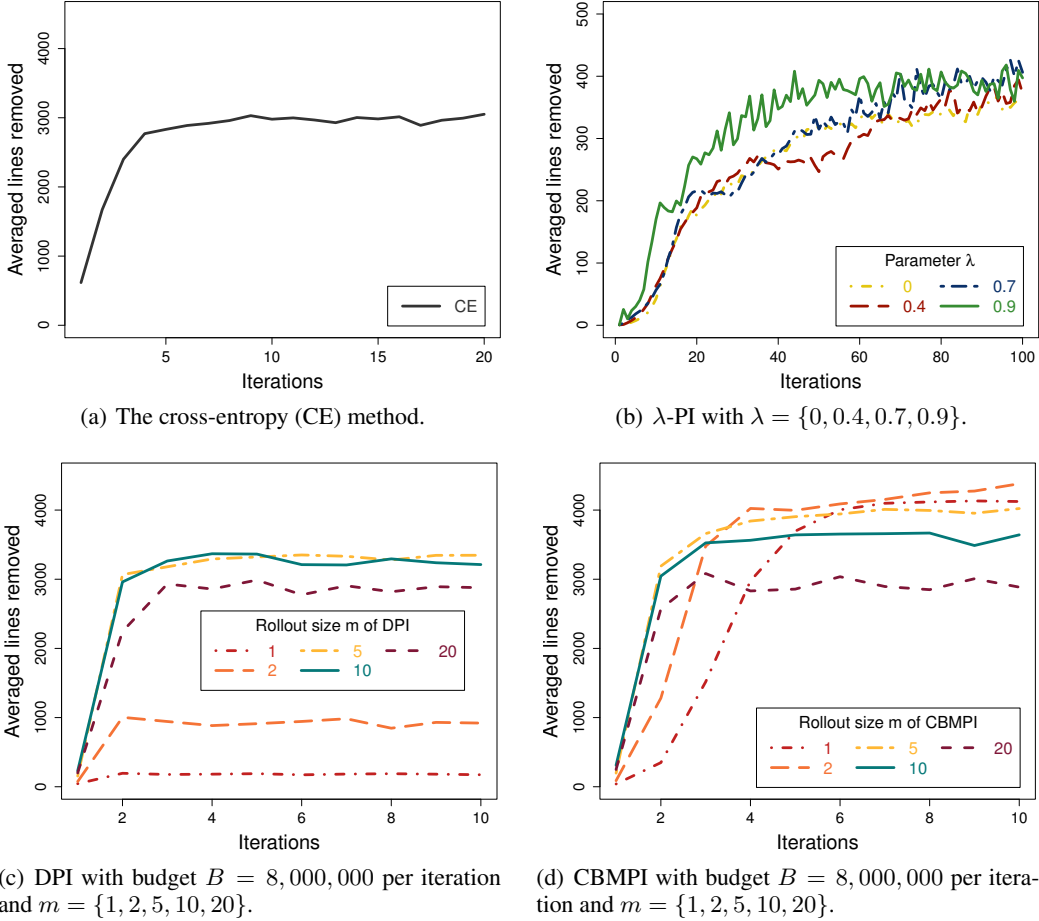

(a) The cross-entropy (CE) method.

(b) $\lambda$-PI with $\lambda = \{0, 0.4, 0.7, 0.9\}$.

(c) DPI with budget $B = 8,000,000$ per iteration and $m = \{1, 2, 5, 10, 20\}$.

(d) CBMPI with budget $B = 8,000,000$ per iteration and $m = \{1, 2, 5, 10, 20\}$.

Figure 4: Learning curves of CE, $\lambda$-PI, DPI, and CBMPI algorithms using the 9 Dellacherie-Thiery (D-T) features on the small $10 \times 10$ board. The results are averaged over $100$ runs of the algorithms.

does not lead to a significant performance improvement over DPI. At the end, we tried several values of $m$ in this setting among which $m = 10$ achieved the best performance for both DPI and CBMPI.

### 3.2.2 Large ($10 \times 20$) Board

We now use the best parameters and features in the small board experiments, run CE, DPI, and CBMPI algorithms in the large board, and report their results in Figure 5 (d). The per iteration budget of DPI and CBMPI is set to $B = 16,000,000$. While $\lambda$-PI with per iteration budget $620,000$, at its best, achieves the score of $2500$ (due to space limitation, we do not report these results here), DPI and CBMPI, with $m = 10$, reach the scores of $12,000,000$ and $21,000,000$ after 3 and 6 iterations, respectively. CE matches the performances of CBMPI with the score of $20,000,000$ after $8$ iterations. However, this is achieved with almost 6 times more samples, i.e., after 8 iterations, CBMPI and CE use $256,000,000$ and $1,700,000,000$ samples, respectively.

**Comparison of the best policies:** So far the reported scores for each algorithm was averaged over the policies learned in $100$ separate runs. Here we select the best policies observed in our all experiments and compute their scores more accurately by averaging over $10,000$ games. We then compare these results with the best policies reported in the literature, i.e., DU and BDU [20] in both small and large boards in Table 1. The DT-10 and DT-20 policies, whose weights and features are given in Table 2, are policies learned by CBMPI with D-T features in the small and large boards, respectively. As shown in Table 1, DT-10 removes 5000 lines and outperforms DU, BDU, and DT-20 in the small board. Note that DT-10 is the only policy among these four that has been learned in the small board. In the large board, DT-20 obtains the score of $51,000,000$ and not only outperforms the other three policies, but also achieves the best reported result in the literature (to the best of our knowledge).

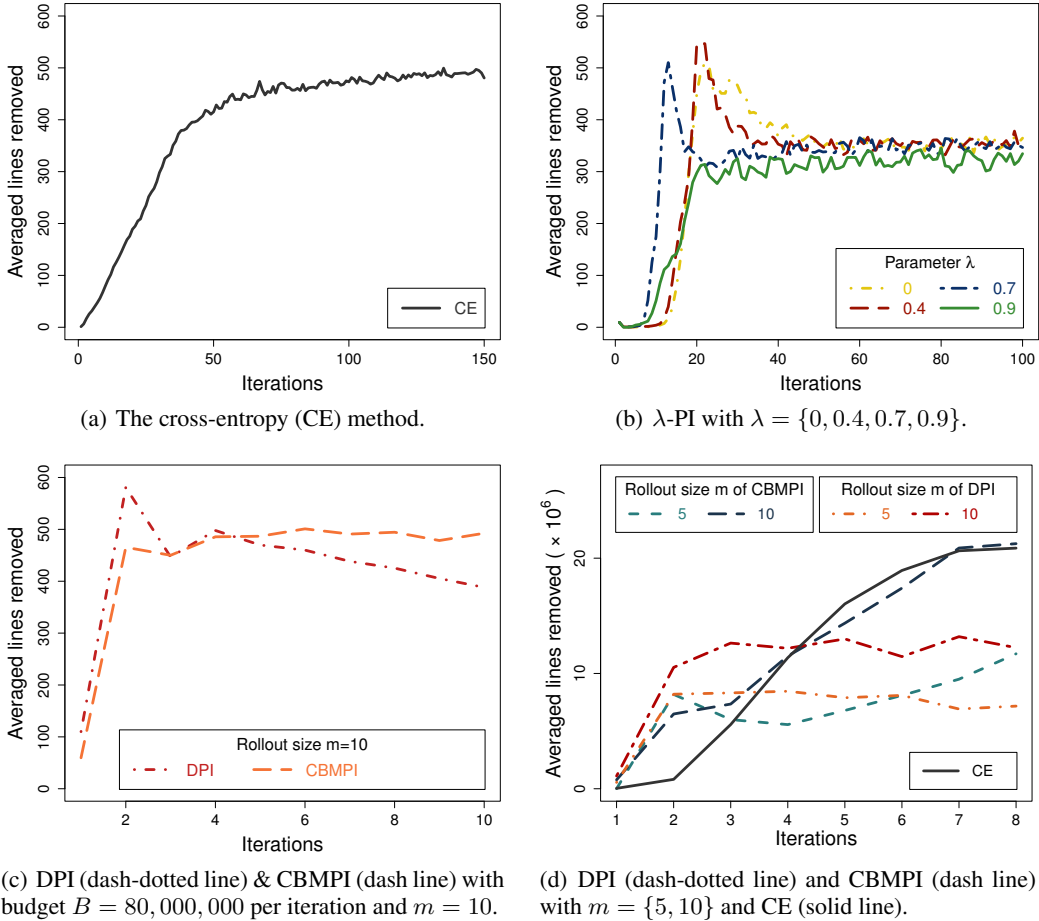

(a) The cross-entropy (CE) method.

(b) $\lambda$-PI with $\lambda = \{0, 0.4, 0.7, 0.9\}$.

(c) DPI (dash-dotted line) & CBMPI (dash line) with budget $B = 80,000,000$ per iteration and $m = 10$.

(d) DPI (dash-dotted line) and CBMPI (dash line) with $m = \{5, 10\}$ and CE (solid line).

Figure 5: (a)-(c) Learning curves of CE, $\lambda$-PI, DPI, and CBMPI algorithms using the 22 Bertsekas features on the small $10 \times 10$ board. (d) Learning curves of CE, DPI, and CBMPI algorithms using the 9 Dellacherie-Thiery (D-T) features on the large $10 \times 20$ board.

| Boards \ Policies | DU | BDU | DT-10 | DT-20 |
|---|---|---|---|---|
| Small ($10 \times 10$) board | 3800 | 4200 | 5000 | 4300 |
| Large ($10 \times 20$) board | $31,000,000$ | $36,000,000$ | $29,000,000$ | $51,000,000$ |

Table 1: Average (over $10,000$ games) score of DU, BDU, DT-10, and DT-20 policies.

| feature | weight | | feature | weight | | feature | weight | |
|---|---|---|---|---|---|---|---|---|
| landing height | -2.18 | -2.68 | column transitions | -3.31 | -6.32 | hole depth | -0.81 | -0.43 |
| eroded piece cells | 2.42 | 1.38 | holes | 0.95 | 2.03 | rows with holes | -9.65 | -9.48 |
| row transitions | -2.17 | -2.41 | board wells | -2.22 | -2.71 | diversity | 1.27 | 0.89 |

Table 2: The weights of the 9 Dellacherie-Thiery features in DT-10 (left) and DT-20 (right) policies.

## 4  Conclusions

The game of Tetris has been always challenging for approximate dynamic programming (ADP) algorithms. Surprisingly, much simpler black box optimization methods, such as cross entropy (CE), have produced controllers far superior to those learned by the ADP algorithms. In this paper, we applied a relatively novel ADP algorithm, called classification-based modified policy iteration (CBMPI), to Tetris. Our results showed that for the first time an ADP algorithm (CBMPI) performed extremely well in both small $10 \times 10$ and large $10 \times 20$ boards and achieved performance either better (in the small board) or equal with considerably fewer samples (in the large board) than the state-of-the-art CE methods. In particular, the best policy learned by CBMPI obtained the performance of $51,000,000$ lines on average, a new record in the large board of Tetris.

## Footnotes

[2]The total number of actions at a state depends on the falling piece, with the maximum of 32, i.e. $|\mathcal{A}| \leq 32$.

[3] Note that the function space $\mathcal{F}$ and policy space $\Pi$ are defined by the choice of the regressor and classifier.

[4]For a precise definition of the features, see [19] or the documentation of their code [21].

[5]Note that we use D-T+5 features only for the value function of CBMPI, and thus, we have a fair comparison between CBMPI and DPI. To have a fair comparison with $\lambda$-PI, we ran this algorithm with D-T+5 features, and it only raised its performance to $800$, still far from the CBMPI's performance.

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
