[Reviews · NeurIPS 2013]

Submitted by Assigned_Reviewer_7

The paper empirically evaluates classification based policy iteration (CBPI) on the Tetris benchmark problem. There are no new technical results; instead the authors analyze the impact of various parameters of the algorithm on its performance.

The results show that direct policy iteration (CBPI that does not use value functions) performs essentially as well as the cross-entropy method and classification-based policy iteration. In some sense, I do not find this too surprising since the DPI is very similar to the cross-entropy method in the way that policies are represented and optimized.

I do find the analysis and comparison convincing in that the standard Tetris features are not suitable for representing value functions optimized by approximate policy iteration. However, the paper comes short of identifying why this may be the case or how one can design better features. This limits the importance of the results and their implication for other problems. In addition, I would like to see included other algorithms that compute approximate value functions in a different way from policy iteration. As it stands, the results show that good value functions cannot be computed using policy iteration, but that does not mean that other algorithms cannot find a good representation. For example, Smoothed Approximate Linear Programming [Desai, Farias, et al, 2012] has been used to obtain moderately encouraging results on this test domain and should be included in the comparison.
Summary: The paper presents thorough evaluation of some major algorithms on Tetris, which is an important RL benchmark problem. While the evaluation shows that classification-based policy iteration performs better than DP, the paper lacks additional insights into the reasons for the difference in solution quality which limits the applicability of the results to other (practical) domains.

Submitted by Assigned_Reviewer_8

This paper studies the application of a classifier-based approximate policy iteration algorithm called CBMPI to the game of Tetris, a much studied benchmark task in reinforcement learning. CBMPI is compared to DPI, a variant which omits the regression step used to estimate the value function in CBMPI; lambda-PI, a related policy iteration method which uses eligibility-trace-like backups and omits the classification step of CPMPI; and CE, a policy-search method that is currently the state of the art in Tetris. Experiments are presented on small boards and large boards using different state features, and the best policies across runs for each method are further evaluated across 10,000 games.

Though this paper makes no theoretical or algorithmic contribution, the empirical analysis is potentially highly significant. Because this benchmark task is well studied, significantly improving the state of the art is a nontrivial endeavor. In addition, since the state-of-the-art CE approach has high sample complexity, progress getting strong performance from more sample-efficient value-function methods is of substantial interest, especially if the results shed light on *why* previous efforts with value-function methods performed poorly.

However, I have two main concerns. First, I find the paper's efforts to address the "why" unsatisfactory. Second, the experimental comparisons are plagued by confounding factors that substantially undermine the validity of the conclusions drawn from them.

Regarding the "why", the authors propose that the success of CE suggests that in Tetris, policies may be easier to represent than value functions. That's a reasonable hypothesis, and it is consistent with the folk wisdom that good policies are sometimes simpler than accurate value functions, and that this can give policy search an advantage. However, the authors then suggest that, if this hypothesis holds, a consequence thereof would be that we should employ ADP methods that search in policy space rather than value function space, and they propose CBMPI as a candidate. However, this is essentially an actor-critic method that explicitly represents both the value function and the policy and thus the difficulties of representing a value function have in no way been circumvented. (This is of course true of all ADP methods because if they didn't explicitly represent the value function they would by definition be policy-search methods). It's true that CBMPI conducts a search in policy space for a classifier-based policy that is consistent with estimated Q-values, and that this is different from many ADP methods, but we are never given any argument why this should make Tetris easier to solve and it does not follow from the hypothesis that Tetris policies are easier to represent than value functions that this would be the case. The superior performance of CBMPI to lambda-PI, which does not explicitly search in policy space, would seem to lend credence to the authors' claim (though without addressing the "why") but there are confounding factors in these comparisons (see below) that leave me unconvinced.

Regarding the confounding factors, I have the following concerns about the experiments:

1. In 2.2, we are told that CBMPI depends on rollouts starting from a set of states sampled from a distribution mu. This already raises questions about comparing to CE, because CBMPI requires a generative model whereas a CE requires only a trajectory model. Then, in 3.1, we learn that the set of states are not sampled from a stationary mu but from the trajectories generated by a strong Tetris policy DU which is assumed to be available as prior knowledge. However, in Appendix A, we are told that lambda-PI was deprived of this prior knowledge and instead used a Dirac on the empty board state. In addition, CE was also deprived of this prior knowledge. Though it is less obvious how CE should make use of this initial policy, one could certainly imagining seeding the initial Gaussian distribution on the basis of it. In my opinion, this raises serious questions about the fairness of the comparisons between CBMPI and lambda-PI and CE.

2. In 3.2.1, we are told that in the small-board experiments using the D-T features, CBMPI was optimized for the feature settings and the best performance was obtained using D-T plus the 5 RBF height features. While the paper is not 100% clear on this matter, as far as I can tell, the other methods used only the D-T features, making the comparisons in Figure 4 potentially very unfair. In Figure 5a-c, where all methods use the same Bertsekas features, we see no performance improvement for CBMPI. In Figure 5d, we again see some performance improvement but here again, though the paper is not explicit about it, as far as I can tell CBMPI is using D-T+5 while the other methods are using only D-T.

3. Also regarding 5d: while it's true that CBMPI does better than DPI after 4 iterations, it's also true that DPI does better from 2-4 iterations. It's awkward to argue that final performance is what matters (and therefore CBMPI is better than DPI) given that CE outperforms both in the long run. It's fair to say that CBMPI learns better than CE in a short term, but it's misleading to conclude that CBMPI achieved equal performance with fewer samples, because this is true at only specific points on the learning curve. CBMPI is faster but does not equal CE's final performance.

4. Because a few random events can mean the difference between a game that ends quickly and one that lasts hours, Tetris is notorious for having high variance in scores for a given policy. However, none of the experimental results in this paper include any variance analysis, so we have no way of assessing the statistical significance of the performance differences observed in Figures 4 and 5. I'm particularly concerned about the differences presented in Table 1, since I suspect that, for policies that can clear millions of lines per game, the cross-game variance is probably huge.

Minor comment: Figures 4 and 5 would be much easier to decipher if the x-axis was samples, not iterations, thereby making it directly possible to fairly compare the methods.







Summary: This paper does not make a theoretical or algorithmic contribution but presents strong results for value-function methods on Tetris, an important reinforcement-learning benchmark on which to date only policy-search methods have done well. However, the authors' explanation for why this result occurs is unsatisfying and the empirical comparisons are plagued by confounding factors that render the conclusions unconvincing.

Submitted by Assigned_Reviewer_9

The paper describes the first RL controller in the literature that has comparable performance to black-box optimization techniques in the game of Tetris. The topic is relevant to the RL community, as Tetris has always been a difficult benchmark domain for RL algorithms.

Quality
The contribution of the paper is to apply a previously published algorithm (CBMPI) for solving the game of Tetris. The authors present competitive results with the previous state of the art optimization algorithm for Tetris - the Cross-Entropy method (CE). The empirical results are convincing and the boost in performance as compared to previous attempts to apply RL controllers to this domain is remarkable.

That being said, the paper lacks a detailed discussion regarding the (possible) reasons for this improvement. While the authors mention in the introduction that the key conjecture (that was positively tested by the paper) is that one should use techniques that search in the policy space instead of the value function space, more details would improve the paper. To be concrete, here are two examples of topics:

1. In lines 252-253 the authors describe the state distribution used for sampling rollout states. This state distribution comes from executing a very good Tetris controller. It would be interesting to describe how strongly this choice of distribution influenced the performance of CBMPI. If the authors tried other solutions that didn't work well, reporting it would be useful for anybody trying to replicate the results. For example, it would be interesting to know whether the states CBMPI visits while learning constitute a reasonable rollout distribution.

2. The optimization algorithm used for the "Classifier" stage of CBMPI is CMA-ES. As discussed in "Path Integral Policy Improvement with Covariance Matrix Adaptation" (ICML 2012) for example, CMA-ES is part of the same family of algorithms as CE. It is thus interesting that an ADP algorithm that uses a CE-type algorithm in its "inner loop" becomes comparable to CE as applied to searching directly in the policy space. So one natural question is: what would happen if CMA-ES would be replaced by other types of optimizers (gradient-based for instance).

Clarity
The paper is well written, easy to follow and the contributions are clearly stated.
I have several minor comments:
- The definition of the "Classifier" in line 222 is confusing: shouldn't the output be "labels" and not differences of Q values?
- In line 266/267, CMA-ES is named "classifier" - but it is optimization algorithm.
- It's not clear why graphs 4a and 4d are disjoint but 5d contains the performance comparison for all algorithms.
- Section 3.1 - "Experimental" is spelled incorrectly.

Originality and Significance
The paper solves a difficult problem in a novel way and provides interesting insights about how to apply RL algorithms to non-toy domains. It doesn't introduce new algorithmic or theoretical tools. As discussed above, I think the results are significant for the RL community.

Finally, I have several questions:
1. If the paper is accepted, are the authors willing to share the code for reproducing their experiments?
2. The variance of scores tends to be high in Tetris so the graphs should be modified to include confidence intervals for the mean values. So, are the differences we see in the graphs (Figure 4D for example) statistically relevant?
3. Regarding the claim in the sentence in lines 393-395, what happens if CBMPI receives the same number of samples as CE? Will it improve, or is the learning curve already at a plateau so more samples wouldn't help?
4. Regarding the comparison of the best policies - how did you compute the scores of the policies DU and BDU? Did you take the values from the referred paper ([20]), or did you execute those policies using your code base? I'm wondering what is the performance of the best policy discovered by CE in your experiments, and I wanted to make sure such a result is reported.
Summary: The paper needs a more detailed discussion of the results to maximize its impact and usefulness to the community. But it is a valuable contribution to the literature that sheds new light on a previously unknown phenomenon: the poor performance of RL algorithms in the game Tetris.
Author Feedback

Author rebuttal: We'd like to thank the reviewers for their useful comments.


"CBMPI uses same number of samples as CE" (Reviewers 8 & 9)
What we currently have in the paper is that CBMPI achieves a slightly worse performance than CE with almost 13 times less samples. After the NIPS deadline, we ran new experiments in which we increased the number of samples in CBMPI and we managed to achieve the same performance as CE (even a bit better) with 6 times less number of samples.


"DPI is very similar to CE" (Revs. 7 & 9)
It is true that the classifier of DPI and CBMPI belongs to the CE family, and plays an important role in the good performance of DPI (CBMPI). However, what is important is that DPI (CBMPI) achieves the same performance as CE with much less number of samples, and this comes from the ADP nature of these algs.
To respond to Rev. 9, we are currently experimenting with SVM-type of classifiers (in place of classifiers in the CE family) in DPI (CBMPI), but we still do not have results to report.


"states are sampled from a good policy" (Revs. 8 & 9)
As mentioned in the paper (lines 270-273), using the samples from a good policy in CBMPI does not in fact give us a good performance, because it is biased towards boards with small height. We obtain good performance when we use boards with different heights, basically making the sampling distribution more uniform. This is in fact consistent with what the CBMPI (DPI) performance bounds tell us (see the discussion on the concentrability coefficients in these bounds). However, it should be mentioned that the choice of the sampling distribution is an important open question in batch RL, and could have significant effect in the performance of these algs.
To respond to Rev. 8, CBMPI only samples from the good policy and does not need to know its parameters. Basically it is using it as a black box. As a black box, the good policy cannot help CE, it can only help CE if we give it the parameters of this policy. So, CE has not been deprived of any prior knowledge in our experiments.


"confidence intervals" (Revs. 8 & 9)
We will add 98%-confidence interval to our graphs in the final version of the paper. Just to give the Revs. an indication, their maximal range is of 300 lines in the small board and 3,000,000 lines in the large board around its mean, which are respectively about 3500 and 22,000,000 lines. Those ranges are even 3 times smaller in Table 1.


"why value function methods performed poorly" (Revs. 7 & 8)
Based on the existing results, there is an agreement in the field that Tetris is a problem in which the policies are easier to represent than value functions. This means: with the features that immediately come to mind (not very complicated ones that have no intuitive meaning) or reasonable number of features, we can represent good policies but not good value functions. In fact, Szita showed that with the features commonly used in Tetris, we cannot even learn the value function of a very good policy found by CE. The goal of this paper is not to show how to design better features for value function approximation in Tetris. Feature selection is a difficult problem in ML. The goal of this work is to show that this specific class of ADP algs. (classification-based PI) that search in the space of polices, unlike the more traditional value function based ADPs, are capable of learning good policies in Tetris. We also show that we can achieve this goal with less number of samples than the direct policy search methods.



- Rev 7

"Comparison with other ADP algs."
In Introduction, we reported the existing results of applying ADP algs. (other than PI) such as approximate value iteration and approximate linear programming to Tetris. Since these algs. have not performed well in this game, we did not include them in our experiments. We only compared CBMPI and DPI with the methods that are considered state of the art in Tetris, like CE and lambda-PI (which is an alg. between policy and value iteration). As the reviewer also agrees, SALP (Desai 2012) is not among the algs. that have obtained outstanding results in Tetris.



- Rev 8

"CBMPI is an actor-critic"
It is true that CBMPI is an actor-critic alg. However, what is important is the good performance of CBMPI comes from its classifier, and not its value function approximation. This is why DPI has similar performance to CBMPI in our experiments. Value function only slightly improves the performance (over DPI) by doing some sort of bias-variance tradeoff like all actor-critic algs. So, this is consistent with our suggestion that we shall use ADP methods that search in the policy space rather than value function space in Tetris.


"D-T+5 features"
The comparison in Fig. 4 is not "very unfair" as claimed by the reviewer. We use D-T features for DPI, lambda-PI, and the classifier of CBMPI, and D-T+5 for the value function of CBMPI. So, the comparison between DPI and CBMPI is totally fair, because their classifiers are using the same set of features. We agree with the reviewer that the comparison between lambda-PI and CBMPI might not be fair, because the former uses D-T and the critic of the latter uses D-T+5. However, this does not affect what we conclude from Fig. 4, because the performance of lambda-PI is about 400, while the performance of CBMPI is about 4000. We in fact tried lambda-PI with D-T+5 and it only slightly improves its performance. We shall report this result (lambda-PI with D-T+5) instead of what we currently have in Fig. 4(b) in the final version of the paper.



- Rev 9

"share the code"
We will be very happy to share our code with other researchers. In fact, we have already given it to another group.


"computing the scores of DU & BDU"
We computed the scores of DU and BDU by running them 10000 times using our code. The best performance of CE in our experiments was 34 million lines. We will include this in the final version of the paper.